# Transforming Palliative Care for Rural Patients with COPD Through Nurse-Led Models

**DOI:** 10.3390/healthcare13141687

**Published:** 2025-07-14

**Authors:** Kristen Poston, Alexa Nasti, Carrie Cormack, Sarah N. Miller, Kathleen Oare Lindell

**Affiliations:** College of Nursing, Medical University of South Carolina, 99 Jonathan Lucas Street, MSC 160, Charleston, SC 29245, USA; mcdanikf@musc.edu (K.P.); nasti@musc.edu (A.N.); cormackc@musc.edu (C.C.); millesar@musc.edu (S.N.M.)

**Keywords:** nurse-led, palliative care, education, chronic obstructive pulmonary disease (COPD), rural

## Abstract

**Background/Objectives**: Chronic obstructive pulmonary disease (COPD) is a leading cause of morbidity and mortality worldwide, with rural populations experiencing higher prevalence and worse outcomes. This paper explores the transformative potential of nurse-led palliative care models in addressing the unique challenges faced by rural patients with COPD and their informal caregivers and synthesizes current evidence on nurse-led palliative care interventions, highlighting their impact on symptom management, advance care planning, and psychosocial support. **Methods**: This is a comprehensive synthesis of nurse-led palliative care programs, focusing on home-based care, telehealth, community outreach, and primary care integration. **Results**: Nurse-led interventions significantly improve patient satisfaction, reduce symptom burden, and enhance the likelihood of advance care planning discussions. Home-based care models and telehealth are particularly effective in rural settings, offering accessible and continuous support. **Conclusions**: Nurses have a critical role in bridging the palliative care gap for rural patients with COPD and their informal caregivers. Expanding nurse-led palliative care services can improve quality of life, reduce healthcare utilization, and promote health equity. Future research should focus on long-term outcomes, cost-effectiveness, and strategies for scaling nurse-led palliative care programs in rural contexts.

## 1. Introduction

Chronic obstructive pulmonary disease (COPD), the third leading cause of death in the world, is a progressive respiratory disease that presents a considerable burden and has a significant impact on quality of life (QOL) [1]. The burden of COPD is higher in rural areas with higher COPD prevalence and worse outcomes, with rural communities experiencing nearly double the COPD prevalence compared to their urban counterparts [2,3]. Patients experience debilitating symptoms such as dyspnea, chronic cough, and fatigue that can severely affect their daily functioning and overall well-being [4]. Patients often avoid activities that may trigger symptoms of dyspnea, one of the most distressing COPD symptoms, contributing to social isolation and overall worse QOL in the COPD population [5,6,7,8].

National guidelines advocate for the provision of palliative care to patients with serious illnesses like COPD [9]. Palliative care offers a comprehensive approach to managing life-limiting symptoms and improving quality of life for both patients and their informal caregivers [10]. Early delivery of palliative care is known to *improve* symptom control, *limit* aggressive end-of-life (EOL) care not aligned with patients’ wishes, *decrease* caregiver (CG) stress, and improve quality of life for the both the patient and CG [11]. Despite these benefits, palliative care remains significantly underutilized in COPD [4,12,13]. Compared to those with lung cancer and other pulmonary diseases, patients with COPD experience the largest palliative care gap and highest hospitalizations, intensive care use, and in-hospital deaths [14]. Concerningly, patients with COPD from rural communities often face greater challenges in accessing palliative care [13,15]. These barriers include too few specialists, lack of PC knowledge, disease uncertainty, and inequitable access to care by geographic region [15]. The immense burden of living with advanced COPD is magnified in the absence of palliative care support.

Nurses can address the palliative care gap, as they play a critical role in delivering palliative care to rural patients with COPD, often serving as primary care providers. Nurse-led interventions have emerged as an effective strategy for bridging gaps in access and quality of care [16]. These interventions often include symptom management, advanced care planning, and psychosocial support and may be delivered using diverse modalities such as home visits, telehealth, and community-based programs. Nurses frequently implement these interventions independently or as part of interprofessional teams, often coordinating closely with physician partners to ensure continuity and alignment of goals. Nurses’ consistent presence and holistic approach to care allow them to recognize changes in patient status and proactively address patient and family concerns, reinforcing their role in various care models globally. However, there remains a need to explore the formal training and preparation of nurses to provide this specialized care. The purpose of this paper is to synthesize the current evidence on nurse-led palliative care interventions for patients with COPD in rural settings and to identify key gaps in nursing education and training to make recommendations for palliative care education for nursing students caring for patients with serious respiratory illness.

### Methodological Approach

This synthesis was guided by expert review and thematic analysis of the existing literature to identify nurse-led palliative care interventions for patients with COPD in rural settings. Expert reviewers included nurse scientists and clinicians with clinical and academic expertise in palliative care, COPD management, rural health, and nurse-led interventions. The team collaboratively defined the focus of inquiry and search terms as follows: “COPD,” “rural” setting, “telehealth,” “nurse-led interventions,” and “palliative care.” A targeted search strategy was developed and implemented using the PubMed, CINAHL, and Scopus databases and supplemented by manual searches of reference lists and key guidelines. Clinical guidelines and program evaluations were also reviewed. Articles published in English within the last 15 years were included in the search. The selection of literature was guided by relevance to the predefined focus areas and the applicability of the findings to nurse-led models of care. Interventions were included if they addressed the role of nurses in delivering palliative care services to patients with COPD and other chronic lung diseases in a particular rural context. A visual overview of the selection process is depicted in Figure 1. Key elements extracted from the evidence included author, year, intervention type, care delivery modality, patient population, outcomes related to palliative care, and nursing considerations. The core characteristics of the interventions are summarized in Table 1. Data synthesis involved thematic grouping of findings across sources to identify effective strategies, common barriers, and gaps in education and training. All study team members participated in the search process and reached consensus on findings to facilitate a comprehensive, practice-informed synthesis of emerging evidence to guide future research and inform development of nurse-led palliative models in rural COPD populations.

## 2. Nurse-Led Palliative Care Interventions for Patients with COPD

### 2.1. Rural Disparities in COPD Care

Rural patients living with COPD face unique challenges in accessing palliative care [24]. Individuals in rural communities experience a higher prevalence of COPD, often due to increased exposure to environmental pollutants (e.g., coal, pesticides, livestock and agriculture exposures, burning biomass, and wood smoke) and limited healthcare resources [25,26,27]. In addition, loneliness impacts these patients with COPD, who are socially isolated, including many who are “tethered” to their supplemental oxygen [28]. Due to low population density across a wider geographic region, rural patients have access to fewer healthcare providers and may face transportation challenges, which hinder their ability to access medical care [24]. This contributes to delayed treatment, inadequate symptom management, and ultimately poor health outcomes. Many rural healthcare facilities lack pulmonologists or palliative care teams, as well as other specialties. This forces patients to rely on general practitioners who may have limited expertise in palliative care medicine, particularly in patients with COPD. A misconception also still exists, in that palliative care and end-of-life care are terms that are used interchangeably. However, palliative care should be provided across a client’s entire disease process to maximize quality of life, and not just in the end stages leading to death to make the most of precious time.

### 2.2. Nurse-Led Interventions in Rural Settings for Palliative Care Patients with COPD

Nurse-led interventions focus on symptom management (e.g., dyspnea, chronic cough), advance care planning, and psychosocial support [4]. Nurse-led care improves patient satisfaction, reduces symptom burden, decreases cost, and increases the likelihood of advance care planning discussions [29]. Integrating palliative care principles, nurses help guide end-of-life discussions, ensuring that patient preferences are considered by the entire team and that the care that is provided is aligned with their wishes and goals. Advance care planning, delivered via nurses, contributes to an improvement in end-of-life discussions for patients with COPD [30].

### 2.3. Home-Based Care Models

Home-based palliative care programs are crucial in rural settings, as they allow patients to receive care in their home rather than traveling long distances for medical services. Advanced practice nurses (APNs) and registered nurses (RNs) can effectively lead home-based programs, providing comprehensive symptom management, advanced care planning discussions, and psychosocial support. Home-based palliative care helps prevent unnecessary hospitalizations and ensures continuity of care by coordinating with primary care providers and specialists. These findings align with broader research indicating that early integration of palliative care in COPD management can lead to better quality of life and reduced healthcare utilization. For instance, a review in CHEST highlights that early palliative care is appropriate at any point during the COPD trajectory, emphasizing the provision of comprehensive support for patients and their care partners [1]. A strong support system is important for patient satisfaction and implementation of palliative care [31]. Incorporating home-based palliative care, especially in rural settings, enhances support and addresses barriers such as limited access to specialized care and transportation challenges. Nurse-led interventions are particularly effective in these contexts, offering tailored care that meets the unique needs of rural patients with COPD. Overall, the evidence strongly supports the expansion of home-based, nurse-led palliative care services to improve outcomes and reduce healthcare burdens for patients with COPD in underserved rural areas. A notable pilot study on nurse-led home-based palliative care in rural communities involved 25 older adults and 11 of their family members living with advanced chronic illness in rural communities. Participants received bi-weekly home visits from a nurse navigator who provided symptom management, education, advance care planning, advocacy, and psychosocial support. Study findings indicated high satisfaction with the service, minimal emergency room use, and a high rate of patients dying in their preferred place, underscoring the feasibility and effectiveness of nurse-led palliative care in rural settings [17].

### 2.4. Telehealth

Telehealth and remote patient monitoring have emerged as transformative tools in rural palliative care. Nurse-led telehealth programs allow for real-time symptom assessment, medication management, and virtual consultations, reducing the burden of travel for rural patients. Through video consultations and mobile health applications, nurses can provide ongoing patient education, facilitate adherence to medication regimens, and address concerns in a timely manner [27]. In one nurse-led intervention delivered via telehealth for patients with COPD, early palliative care was found to be feasible and acceptable to patients with COPD and their caregivers [18,19]. Remote monitoring devices track health indicators such as dyspnea, oxygen saturation, respiratory rate, and heart rate, allowing nurses provide early intervention when deterioration occurs [32]. By integrating telehealth into routine care practices, nurses can offer continuous support, foster patient engagement, and ensure that patients with COPD receive timely, high-quality palliative care despite geographical barriers. Another study found that telehealth significantly improves access to palliative care by reducing the need for extensive travel, enhances symptom management through remote monitoring, and increases patient and caregiver satisfaction [33]. The use of telehealth also minimizes expenses associated with in-person clinic visits and travel costs/time, which may be burdensome to this group. The ability to provide care remotely also enhances accessibility for those with limited mobility or lack of transportation, ultimately promoting equitable healthcare access in underserved regions [33]. However, limited internet access, which can be due to the infrastructure or financial limitations, may affect the implementation of telehealth solutions.

### 2.5. Community Health Programs/Outreach

Community health nursing and outreach programs play an essential role in addressing social determinants of health that affect rural patients with COPD. Mobile health clinics staffed by nurses can bring essential palliative care and other medical services directly to underserved areas, offering in-home assessments, education on COPD self-management, and assistance in overcoming transportation and financial barriers. Nurses in community outreach programs foster trust within rural communities by engaging in culturally competent care and providing group-based interventions for peer support. A study by Moy et al. [20] demonstrated that internet-mediated and mobile health interventions, such as a pedometer-based walking program for patients with COPD, significantly improved health-related quality of life and engagement in self-care activities. Participants of this study were veterans with COPD who were randomly assigned to either use the online walking program—which included daily step goals, feedback, education, and community support via an online forum—or be part of a control group that did not participate. Over 12 months, those in the program significantly increased their daily step counts and showed improvements in quality-of-life measures, suggesting that remote, tech-supported exercise programs can be effective for managing COPD. This also highlighted the effectiveness of community initiatives in rural settings, which can be nurse-led in collaboration with the patient’s provider. Such groups can also provide patients with a sense of connection, and that they are not facing this disease alone.

### 2.6. Primary Care in Rural Communities

Integrating palliative care into primary care settings is another effective nursing-led intervention. Given that many rural patients with COPD rely on primary care as their primary healthcare access point, embedding palliative care services into these settings ensures early and ongoing support. Nurses educated in palliative care, during pre-licensure training or by additional certification, can train primary care providers in palliative care principles, lead goals-of-care discussions, and advocate for palliative care referrals when appropriate. By doing so, nurses help create a seamless continuum of care, ensuring that symptom management, psychological support, and advanced care planning are proactively addressed. The nursing profession is flexible and offers the ability to not only work as a clinical professional, but also serve as a navigator, guiding patients to the appropriate resources. This approach reduces symptom burden, enhances quality of life, and aligns patient care with their preferences [34]. A study in Australia found that embedding nurse-led supportive care in an existing outpatient COPD service enhanced the primary palliative care skills of respiratory specialist clinicians, resulting in more successful collaboration between both the respiratory and palliative care services, and recommended further study from the perspective of the patients and families [21].

In rural areas of the United States, primary care providers can be scarce and cover a wide range of territory with a high patient load. Nurse practitioners (NPs) help bridge the gap in rural healthcare by providing comprehensive primary care services in areas with few or no physicians, often serving as the main point of care for many residents. Palliative care nurse practitioners (NPs) and nurses play a vital role in addressing the complex healthcare needs of patients in rural areas, where access to specialized care is often limited. These nurse practitioners can act as either a primary care provider or a specialist (or both in some instances) and provide essential services such as dyspnea and symptom management, medication education, and psychosocial support, helping patients with serious or life-limiting illnesses maintain quality of life close to home. With their expertise in holistic, patient-centered care, palliative care NPs often serve as the primary source of support for both patients and families, particularly in areas lacking hospice or specialty services [35]. Advanced practice nurses and nurse practitioners possess the ability to coordinate care across multiple settings—including home visits, telehealth, and local clinics, which reduces hospitalizations and emergency department use that are often burdensome for rural residents who face long travel distances.

### 2.7. Nurse-Led Interventions for Advance Care Planning and Cost Effectiveness

Nurses play a crucial role in facilitating advanced care planning (ACP) and providing education to patients with COPD and their informal caregivers, aiming to improve health outcomes and reduce the financial burdens associated with hospital readmissions and the equipment required for oxygen therapy. Nurses not only initiate these conversations, but also provide continuous guidance, ensuring that patients and their informal caregivers understand disease progression, potential complications, and palliative care options. A study in the Netherlands [22] demonstrated that a single, structured, nurse-led ACP session significantly improved the quality of end-of-life care communication between patients with COPD and their physicians. This intervention not only facilitated meaningful discussions about treatment preferences and end-of-life options, but also did so without causing psychological distress to patients or their loved ones. Notably, the study found that loved ones of patients who participated in the ACP session experienced reduced anxiety levels. These findings underscore the pivotal role nurses play in initiating and guiding ACP discussions, ultimately enhancing patient autonomy and preparedness for end-of-life decisions.

In rural and resource-limited settings, optimizing COPD management with low-cost interventions can significantly reduce patients’ symptoms, as well as their reliance on home oxygen treatment. This not only reduces healthcare costs, but also improves the patient’s quality of life. At-home pulmonary rehabilitation, including guided walking routines and breathing exercises, has been shown to improve lung function and overall quality of life in COPD patients. Inhaler education by RNs is critical, as studies reveal that a large proportion of patients misuse their inhalers, leading to suboptimal drug delivery and increased symptom burden [36]. Additionally, nutritional support is vital, as malnutrition exacerbates respiratory muscle weakness, increasing dyspnea and hospitalizations [37]. Both advanced practice nurses (NPs) and registered nurses (RNs) are equipped to provide all this education to their patients. Finally, smoking cessation remains one of the most impactful and cost-effective strategies in COPD management, drastically improving lung function and slowing disease progression [38]. Advanced practice nurses and registered nurses can provide guidance and become certified as Tobacco Treatment Specialists to provide comprehensive individualized guidance on how a patient can quit smoking. Collectively, these interventions offer an affordable, sustainable framework for managing COPD in underserved areas.

Nurses can also initiate cost-effective home-based equipment to reduce the financial burdens patients with COPD or their caregivers may be experiencing. Reducing the financial burden allows caregivers to focus more on enhancing the patient’s quality of life. An example is long-term, continuous, or intermittent oxygen therapy that patients with COPD may use for treatment of hypoxemia and symptoms of dyspnea. In rural settings, patients may be affected by both their geographical isolation and financial constraints, which can make the delivery of care physically challenging and economically burdensome. An oxygen-conserving reservoir is a specialized oxygen-conserving nasal cannula designed to enhance the efficiency of oxygen therapy, particularly in patients with COPD requiring long-term oxygen [39]. Most of these devices have been available for over 30 years; however, they have not been widely implemented. The device incorporates a built-in reservoir that stores oxygen during exhalation and delivers it as a bolus at the start of inhalation, effectively increasing the fraction of inspired oxygen concentration (FiO_2_) at lower flow rates. This makes these devices ideal for use in resource-limited settings where oxygen supplies may be constrained or concentrator access is limited. This device can also increase patient mobility and exercise tolerance, contributing to better quality of life and potentially fewer hospitalizations. A randomized control trial explored 43 patients with severe COPD who used an oxygen-conserving reservoir during pulmonary rehabilitation, and the results showed that the use of the oxygen-conserving device was superior to a traditional nasal cannula for oxygen delivery, particularly in patients with higher oxygen demands [23]. Nurses play a key role in initiating and managing this therapy, as well as educating the provider on a device that is not widely known or utilized. The initial nursing assessment would include the patient’s baseline oxygen needs and suitability for oxygen-conserving devices. Once connected to a prescribed oxygen source, the reservoir can be used at typical flow rates between 1 and 5 L/min. Nurses monitor the patient’s oxygen saturation at rest and during exertion to ensure target oxygen saturation levels are maintained.

### 2.8. Integrating Rural Palliative Care into Nursing Training

Nurses are uniquely positioned to deliver palliative care in rural sections via implementation of nurse-led care delivery models [40], particularly in community-based settings and rural areas, where lived experiences, environmental and economic barriers, and access to resources may vary greatly [39]. Given that care for patients with COPD is often delivered in rural areas and via home-based palliative care, it is critical that nurses are trained to develop competencies specifically in these modalities of nurse-led care delivery. The American Association of Colleges of Nursing (AACN) 2021 update to *The Essentials: Core Competencies for Professional Nursing Education* emphasizes four spheres (settings) of care, including a hospice/palliative care sphere. Additionally, a team of palliative care experts revised competency statements (CARES and G-CARES), which outline educational expectations for nurses, providing guidance for faculty in implementation of content specific to the hospice/palliative/supportive care sphere [41]. These resources serve as a framework to equip nurses in community-based and rural settings to deliver palliative care for COPD patients, including nuances of care coordination to ensure individualized goals of care are met, as well as an emphasis on the role of hospice and palliative care nurses in advocacy [42].

## 3. Case Examples

### 3.1. Embedded RN in Primary Care

Mr. H., a 73-year-old man with GOLD stage 4 COPD and a history of agricultural farming exposure, lives in a medically underserved rural area in the Southeast U.S. He receives care from a small primary care clinic that lacks onsite pulmonary or palliative specialists. Recognizing the complexity of his needs, including chronic breathlessness, medication side effects, and psychosocial distress, an embedded registered nurse (RN) was assigned to coordinate his care.

The RN conducts monthly home assessments and reviews symptom burden using structured symptom checklists. She identifies poorly controlled dyspnea and increasing caregiver strain, prompting her to initiate a telehealth consult with a palliative care NP at a partnering tertiary center. She leads advance care planning conversations, updates his goals-of-care documentation, and coaches Mr. H. and his daughter on managing oxygen therapy and recognizing exacerbation symptoms. As emphasized in *The Essentials* and the CARES competencies, the RN operates within a framework of person-centered, coordinated care that bridges primary and specialty services.

Discussion:

This case highlights the effectiveness of nurse-led care in rural primary care settings, where patients often face severe limitations in accessing specialty palliative care. The RN’s role in symptom monitoring, advance care planning, and family support demonstrates how embedded nurses can reduce hospitalizations, improve alignment of care with patient values, and provide continuity—especially when formal palliative care teams are not geographically available. This model embodies the interprofessional, SDOH-informed competencies outlined in the AACN *Essentials*, positioning RNs as vital facilitators of palliative care integration in rural COPD management.

### 3.2. Telehealth-Delivered Nurse Practitioner Palliative Support

Ms. L., a 67-year-old Latina woman with advanced COPD, lives in a rural agricultural region more than 60 miles from the nearest hospital. Her symptoms—progressive dyspnea, anxiety, and isolation—limit her ability to leave home, and she frequently misses follow-up appointments. She enrolls in a nurse practitioner (NP)-led telehealth palliative care program designed for rural populations.

Using secure video visits every two weeks, the NP assesses her symptom trajectory, makes titrations to her inhaler and anxiety medication regimen, and provides structured education on breathing techniques and energy conservation. The program also includes remote monitoring with a Bluetooth-enabled pulse oximeter, allowing the NP to track trends in oxygen saturation and intervene early in response to decline. When Ms. L. expresses fear about the future and concern about burdening her family, the NP initiates advance care planning and facilitates a consult with a palliative care social worker—all delivered virtually.

Discussion:

This case illustrates how a telehealth model can extend high-quality palliative care into patients’ homes, overcoming barriers like transportation, mobility, and workforce shortages. Consistent with findings in this review and studies such as [17,18,30,31], the program improves access, symptom control, and patient satisfaction. The NP’s holistic role aligns with the updated *Essentials* domains—particularly healthcare technology integration, person-centered communication, and interprofessional coordination—enabling timely, dignified, and equitable palliative care delivery in an otherwise hard-to-reach population.

## 4. Discussion

This review underscores the critical need for nurse-led palliative care interventions to improve COPD management in rural settings. Rural patients face significant disparities due to geographic and economic barriers, limited access to pulmonologists, and increased hospitalizations. Nurse-led models, including home-based programs, telehealth, community outreach, and primary care integration, enhance symptom management, advance care planning, and psychosocial support. Expanding these models can improve patient-centered outcomes and reduce healthcare burden. Nurse-led interventions address key palliative care needs, improving quality of life and access to care. Rural patients with COPD face inequities exacerbated by socioeconomic and geographic factors. Nurses provide essential services, often serving as the primary point of contact in underserved areas. Telehealth extends the reach of these interventions, improving accessibility for geographically isolated patients.

While telehealth in rural communities seems like a straightforward answer to addressing palliative care needs in patients living in rural areas with COPD, implementing telehealth in rural communities faces several key barriers that hinder its widespread adoption. One of the most significant challenges is limited internet and technology infrastructure. These remote areas may lack reliable WiFi or internet access necessary for high-quality virtual care. Additionally, digital literacy remains a concern, as patients, caregivers, and even some providers may not be comfortable using telehealth tools. While the downstream effects of telehealth are cost-effective when in place, the training of nurses and providers in how to use telehealth/virtual platforms would require additional training, and thus increase cost at inception. The cost of required devices and internet services can also be prohibitive for low-income individuals, as well as specialized HIPAA-protecting software that would keep patient information private while conducting phone and video visits. Patients may also be hesitant to even discuss their private health information over these systems, even if the correct privacy measures are in place [33]. Regulatory hurdles, such as inconsistent licensing and reimbursement policies across regions, further complicate implementation; for instance, a provider located in another state cannot bill for a visit across state boundaries without dual licensure. Lastly, cultural and language differences may not be adequately addressed by existing telehealth platforms, limiting their accessibility and effectiveness in diverse rural populations.

An important issue requiring significant policy reform and increased advocacy is the variation in nurse practitioners’ (NPs) scope of practice across states, which limits their ability to provide comprehensive care in underserved rural communities. While many rural states have adopted Full Practice Authority (FPA), allowing NPs to work to the full extent of their training and licensure, others maintain restricted practice laws that curtail their autonomy—particularly in palliative care delivery [43]. While most states have adopted Full Practice Authority for nurse practitioners, in rural states like Montana, Wyoming, and Kansas, there are still states with large pockets of rural communities that still have restrictive practice authority for nurse practitioners. For example, in South Carolina, NPs operate under restricted practice regulations. Although they can order palliative care services and initiate hospice referrals, they are not permitted to independently diagnose a patient as terminally ill without physician oversight. This is a similar issue in other rural states like Missouri, Oklahoma, and Tennessee. These limitations hinder timely access to essential end-of-life care and underscore the need for policy alignment that supports the full utilization of NPs in meeting the complex healthcare needs of rural populations.

Integrating palliative care education into nursing curricula is essential to prepare future nurses. Historically rooted in acute care, nursing education must expand to include palliative and rural care. The AACN *Essentials* emphasize competency-based education aligned with real-world practice settings. Enhancing training through innovative resources ensures students develop clinical judgment, communication, and decision-making skills necessary for effective palliative care [44]. Emerging technologies, such as immersive virtual reality simulation (IVRS), offer promising approaches to competency development. IVRS provides realistic, interactive scenarios that improve confidence and clinical judgment. Academic–practice partnerships enrich curricula by incorporating real-world expertise, ensuring graduates are equipped to deliver high-quality palliative care [45,46]. The End-of-Life Nursing Education Consortium (ELNEC) curriculum further supports competency development, preparing students to meet the needs of patients with COPD [47]. While integrating palliative care into nursing curricula is critical, access to advanced educational tools like immersive virtual reality simulation (IVRS) may be limited, especially in rural or under-resourced academic institutions [48]. These disparities can lead to uneven training experiences in these areas. Additionally, implementing academic–practice partnerships and curricula, like ELNEC, requires institutional support, faculty training, and adequate funding, which may not be readily available in all nursing programs [49,50]. Institutions would need to increase funding for palliative care training and advanced educational tools. Applying for federal, state, and private grants—particularly those focused on rural health or healthcare education—can provide significant financial support. Forming partnerships with nearby healthcare systems, hospice centers, and community organizations can also bring in resources and real-world training opportunities. Engaging alumni and donors is another option through targeted fundraising efforts. Additionally, offering employer-covered continuing education or certification programs as part of a job offer can be a powerful incentive, encouraging nurses to pursue advanced degrees or specialized training.

Despite the promise of nurse-led interventions, implementation barriers remain. Limited funding, workforce shortages, and disparities in telehealth access hinder widespread adoption. Policymakers must prioritize investment in rural healthcare infrastructure and workforce development. Addressing these challenges through systemic changes, such as expanded telehealth access and improved provider training, is essential for equitable palliative care delivery.

## 5. Limitations

While this review provides valuable insights into nurse-led palliative care interventions for rural patients with COPD, several limitations should be acknowledged. First, the synthesis relies on existing literature, which may include studies with varying methodologies and sample sizes, potentially affecting the generalizability of the findings. Additionally, the review focuses primarily on interventions in rural settings, which may not be directly applicable to urban populations or other healthcare contexts. The heterogeneity of rural communities, including differences in healthcare infrastructure, socioeconomic status, and cultural factors, may also influence the effectiveness and implementation of nurse-led palliative care models. Furthermore, the review does not extensively address the long-term outcomes and cost-effectiveness of these interventions, highlighting the need for future research in these areas. Finally, while telehealth and home-based care are promising strategies, barriers such as limited internet access and digital literacy in rural areas may hinder their widespread adoption. Addressing these limitations through targeted research and policy initiatives is essential for optimizing palliative care delivery for rural patients with COPD.

### Future Research Directions

While nurse-led interventions show promise, gaps remain. Future research should assess the long-term impact of palliative care on COPD outcomes, strategies to enhance telehealth engagement, cost-effectiveness, and implementation science approaches for scaling programs in rural contexts. Examining behavioral frameworks could further inform patient-centered care strategies.

## 6. Conclusions

Nurse-led interventions, supported by telehealth and behavioral frameworks, improve quality of life and care for rural patients with COPD. Addressing access barriers and promoting health equity are essential to these efforts. By prioritizing nurse training, investing in telehealth, and implementing evidence-based interventions, healthcare systems can ensure comprehensive, patient-centered palliative care. Expanding nurse-led initiatives, strengthening academic–practice partnerships, and integrating competency-based education will be crucial to advancing palliative care delivery and improving outcomes for patients with COPD in underserved communities.

## Figures and Tables

**Figure 1 healthcare-13-01687-f001:**
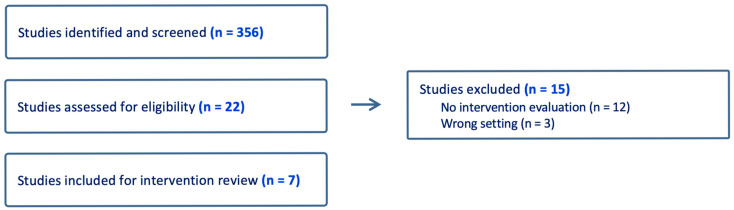
Visual overview of the selection process.

**Table 1 healthcare-13-01687-t001:** Palliative care interventions for patients with COPD in rural areas.

Author (Yr)	Intervention Type	Care Delivery Modality	Patient Population	Palliative Care Outcomes	Considerations for Nursing
Pesut et al. (2017) [17]	Nurse-led home-based palliative care pilot study	In-person, home visits biweekly	Older adults with advanced chronic illness (n = 25) and family members in rural communities (n = 11)	High patient and family satisfaction, minimal emergency room use, many patients died in preferred location when indicated.	Nurse navigators in rural settings can facilitate early palliative care for COPD. Nurses can provide symptom management, education, advance care planning, advocacy, and psychosocial support.
Iyer et al. (2019) [18] Byun et al. (2024) [19]	Nurse-led telehealth palliative care for COPD	Telehealth (video and remote monitoring)	Patients with COPD and their caregivers (n = 60 dyads)	Formative evaluation (Iyer, 2019 [18]) found early PC acceptable, prioritizing coping, symptoms, prognostic awareness, and illness understanding.	Nurses can do real-time assessment, education focused on priority areas, and symptom management and facilitate advance care planning for patients and caregivers.
Moy et al. (2015) [20]	Internet-mediated, mobile health walking program	Mobile health and online platform	Veterans with COPD (n = 239)	Increased daily step counts, improved health-related quality of life.	Nurses can educate patients and facilitate remote, tech-supported exercise programs and foster peer support.
Ora et al. (2023) [21]	Embedded nurse-led supportive care within outpatient COPD service	Outpatient and primary care setting	Patients with COPD in Australia (rural-focused service), semi-structured interviews with healthcare professional (n = 6)	Improved relationships, trust, and collaboration between respiratory and palliative care teams. Benefits for patients, including improved supportive care.	Nurses with clinical expertise can lead models of care that address unmet biopsychosocial and spiritual needs of patients with COPD.
Houben et al. (2019) [22]	A nurse-led structured advance care planning (ACP) session	In-person ACP discussions	Patients with COPD (n = 165)	One 1.5 h structured, nurse-led ACP intervention improved quality of end-of-life communication and reduced anxiety among loved ones.	Nurses can initiate and guide ACP discussions, improving patient autonomy and preparedness, and support emotional well-being in patients and loved ones.
Gloeckl et al. (2014) [23]	Oxygen-conserving nasal cannula use during pulmonary rehabilitation	Pulmonary rehabilitation	Patients with severe COPD (n = 43)	Superior oxygen delivery, improved mobility and exercise tolerance.	Nurses can initiate oxygen-conserving device therapy, monitor patient oxygen levels, and provide education.

## Data Availability

Not applicable.

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
