# Peer review of "Transforming Palliative Care for Rural Patients with COPD Through Nurse-Led Models"

_healthcare, 2025, doi:10.3390/healthcare13141687_

Round 1
Reviewer 1 Report
Comments and Suggestions for Authors
Thpaper is a review focusing on nurse involvement in palliative care specifically for COPD in rural areas. They focus on how nurses can fill the gaps that exist for palliative care,particularly in rural areas, as well as what education is needed to improve their competency.
While it is an important topic and rural patients and those with COPD are particularly populations who do not receive adequate palliative care, I did not find the paper particularly compelling. There were too many generalizations and not enough detail. Details that were interesting were in 2.2 about what the focus of nurse-led symptom management should be (dyspnea, chronic cough, symptoms, ACP, psychosocial support) (lines 84-5). I wanted more specifics, what would nurse-led interventions look like and how would a nurse actually do and how would they interact/liase with physicians and other providers? Instead of just saying as in 2.3 that home based palliative programs with nurses would be beneficial, what would they do exactly? In section 2.4 you give more examples such as with video consultations, remote monitoring and patient education you can facilitate adherence to medications. Also in section 2.5 you talke aout self management assistance can improve health and mobile health clinics. These sections 2.4 and 2.5 were the strongest because they had the most details. It would be helpful to have perhaps case examples to demonstrate how a nurse could improve care for a COPD patient in a rural setting using the various methods, telehealth, mobile clinic, etc. In line 139 you mention pedometer-based walking program, what exactly would a nurse do in relation to the pedometer? For 2.6, it would be helpful to have more details as to how a nurse would train providers on palliative care principals and advocate for referrals? How would they interact with physicians? Specifics on ACP training options for nurses such as Respecting Choices would be helpful. I did not find the section on nurse education (section 3) useful or compelling. I would cancel this section and replace it with case examples and more specific details.
Author Response
Please find the attachment "Response to Reviewers". Thank you.

Reviewer 2 Report
Comments and Suggestions for Authors
The topic of the work is very exciting and highly topical.
Unfortunately, the methodological approach is not very well explained, at least in a few sentences. In my opinion, this would be very relevant for a comprehensive synthesis so that the reader can understand what the authors have done here, as the results seem somewhat arbitrarily chosen and in some cases not well-founded with much literature. Although this is feasible, it does not have a high scientific standard, as it is more assertions. To me, it seems more like a narrative review, which would also have to be described in the procedure. Of course, my impression here may be wrong. Limitations are completely missing here and were not considered as a point. This does not correspond to good scientific practice, but could be quickly rectified by the authors. It would also make transparent what is to be criticised about the methodological approach and that the results cannot be generalised and should not be read as scientifically proven. It would give the article more quality. Some short definitions are also missing. There is a difference between palliative care and end-of-life care, which could be emphasised more (in one sentence).
It could also be emphasised again what exactly Advance Care Planning involves here (at least it could be made clear whether it is, for example, based on ‘Respecting Choices’ [Bud Hemmes et al.] or whether it refers to other areas - it is clear that it is about advance care planning and end-of-life discussions, which is not enough to describe it in my opinion. The discussion has no references and doesn't discuss something from the secondary literatur. For example it says "Policymakers must prioritize investment in rural healthcare infrastructure and workforce development" - this is a conclusion withot any arguments. It would be intresting, what is paid and where is the lack (it is clear, that it is the system of privacy policies - but is there an option in the law about health care to get money from - South Carolina law related to health care includes laws about health insurance, health facility licensing, and patient rights - this is something to look inside to get the argments for a discussion from).
Otherwise, it's a nice writing style and the article is publishable in my view after these corrections.
Author Response

(The authors gave the same response as above.)

Reviewer 3 Report
Comments and Suggestions for Authors
Thank you for the opportunity to review this well-written and informative paper.
A brief summary (one short paragraph) outlining the aim of the paper, its main contributions and strengths: This review paper aims to provide an overview of nurse-led palliative care interventions for those with COPD in rural settings. It provides surface-level information about various palliative care interventions/modalities as well as recommendations for the incorporation of palliative care into nursing education.
General concept comments
Review: commenting on the completeness of the review topic covered, the relevance of the review topic, the gap in knowledge identified, the appropriateness of references, etc.
These comments are focused on the scientific content of the manuscript and should be specific enough for the authors to be able to respond:
- This topic is relevant to nursing in general, nursing education, and those working in palliative care. There is no doubt that the use of palliative care in non-cancer populations is underutilized and is not well-understood in terms of when, where, and how it should be delivered to those in rural communities. This paper is a good overview of the different nurse-led models that have been used to increase access to pall care. The citations are appropriate and recent. However, the evidence related to nurse-led palliative care interventions and their effectiveness in the primary care setting, rural settings, and in this population is generally sparse (compared to say...the oncology world). I recommend adding in some verbiage somewhere that acknowledges this as a limitation.
Specific comments referring to line numbers, tables or figures that point out inaccuracies within the text or sentences that are unclear. These comments should also focus on the scientific content and not on spelling, formatting or English language problems, as these can be addressed at a later stage by our internal staff.
A few comments/suggestions for your consideration:
- The abstract's purpose statement nor the title does not seem to align well with the information in the paper itself. Recommend either editing the objective to incorporate nursing education as a focus or pulling back on the nursing education piece a bit and use it as a major discussion point related to how targeting nursing education will increase pall care competencies which could then in turn positively impact access to nurse-led interventions
- Line 85's reference is not related to COPD - suggest adding in "in other chronic lung populations" and then inferring the possible usefulness in those with COPD to give clarity
- Not sure where the behavioral frameworks recommendation comes from in the conclusion. It doesn't seem to tie into any of the presented interventions in the body of the paper except for a little snippet on self-management in the community-led section
- The education section is missing some citations to support the use of these modalities for assessing competence.
- The discussion is missing a little depth. It seems to just paraphrase the body of the paper. A more robust discussion around limitations in terms of delivery of palliative care and gaps in the literature is warranted here.
Author Response

(The authors gave the same response as above.)

Round 2
Reviewer 1 Report
Comments and Suggestions for Authors
More detail has been added to this paper that is more practical and useful compared to the initial draft. I did not see any hypothetical case examples of patients being treated using nurses. I think 1-2 examples of this would give valuable practical insight into the work and show how nurses can be utilized and what is their value. You could describe how different models of nurse/physician interaction could be beneficial, perhaps one with an NP and one with an RN. One with embedded RNs in primary care and another using telehealth or home visits. I do like how you gave more comprehensive explanations of the studies that you cited, such as the VA pedometer study.
Line 95 – are there actually more environmental pollutants in rural areas? What’s the evidence. Living in Los Angeles with all its smog, I’m not sure this is true, or would ast least I’d like to see evidence.
Lines 236-243 could be cut. It isn’t very useful. Lines 243-256 are great though.
Lines 326-343 I don’t really understand how the virtual simulation reality even really works. Again, a real world example could be very helpful here.
The entire education piece still feel unhelpful and apart from the rest of the work and I would still eliminate it and replace it with several case examples.
Lines 372-386 this part of the discussion feels unrelated. The whole paper is talking about how nurses can be helpful in rural palliative COPD treatment and much of the discussion addresses the barriers to implementation. I think it’s helpful information, but instead of ina discussion I’d rework it in the rest of the paper. Otherwise you start reading the paper on a hopeful note and end with a big bucket of ice water. So talk about the barriers of telehealth when you talk about telehealth.
